Isolation and in silico analysis of a new subclass of parasporin 4 from Bacillus thuringiensis coreanensis

Santos Thais N. F. 1 thais.nayara@unesp.br
http://orcid.org/0000-0002-4253-0320 Moreira Raquel O. 1
Rodrigues Jardel D. B. 1
Rojas Luis A. C. 2
Souza Jackson A. M. 1
Desidério Janete A. 1
1 Biology Department, São Paulo State University , Jaboticabal, São Paulo , Brazil
2 Department of Agricultural and Environmental Biotechnology, São Paulo State University , Jaboticabal, São Paulo , Brazil
Upadhyay Rohit
Electronic publication date: 2025 Mar 24
Publication date: 2025
Volume: 13
Electronic Location ID: e19061
Received 2024 Oct 10; Accepted 2025 Feb 6
Copyright: © 2025 Santos et al.
Copyright year: 2025
Copyright holder: Santos et al.
License: This is an open access article distributed under the terms of the Creative Commons Attribution License, which permits unrestricted use, distribution, reproduction and adaptation in any medium and for any purpose provided that it is properly attributed. For attribution, the original author(s), title, publication source (PeerJ) and either DOI or URL of the article must be cited.
License URL: https://creativecommons.org/licenses/by/4.0/

Keywords: Bacillus thuringiensis, Parasporin, Cry, Cancer, Antitumor activity, Cytotoxicity, Pore former, Bioinformatics, Modeling, Phylogenetics

Funding: Fundação de Amparo à Pesquisa do Estado de São Paulo 2019/21621-0 Coordenação de Aperfeiçoamento de Pessoal de Nível Superior 88887.641450/2021-00 This work was supported by Fundação de Amparo à Pesquisa do Estado de São Paulo (No. 2019/21621-0) and Coordenação de Aperfeiçoamento de Pessoal de Nível Superior (No. 88887.641450/2021-00). The funders had no role in study design, data collection and analysis, decision to publish, or preparation of the manuscript.

==============================
Background

Bacillus thuringiensis (Bt) is a Gram-positive bacterium whose strains have been studied mainly for the control of insect pests, due to the insecticidal capacity of its Cry and Vip proteins. However, recent studies indicate the presence of other proteins with no known insecticidal action. These proteins denominated “parasporins” (PS) have cytotoxic activity and are divided into six classes, namely PS1, PS2, PS3, PS4, PS5, and PS6. Among these, parasporins 4 (PS4) has only one described subclass, present in the Bacillus thuringiensis shandongiensis strain. Given the importance and limited knowledge about the actions of PS4 proteins and the existence of only one described subclass, the present work aimed to characterize the Bacillus thuringiensis coreanensis strain as a potential source of PS4 protein.

Methods

A preliminary screening to detect the ps4 gene was conducted in a bank of standard strains and isolates of Bacillus thuringiensis from the Laboratory of Bacterial Genetics and Applied Biotechnology, FCAV/UNESP. The positive strain for this gene had its genomic DNA extracted, the ps4 gene was isolated, cloned and in silico analyses of its sequence were performed. Tools such as Bioedit, BLAST, Clustal Omega, Geneious, IQ-Tree, and iTOL were used in these analyses. For the structural analysis of the PS4 detected, in comparison to the database PS4 (BAD22577), the tools Alphafold2, Pymol, and InterPro were used. Sodium dodecyl sulfate-polyacrylamide gel electrophoresis (SDS-PAGE) gel analyses allowed the visualization of the inactive and active PS4 protein from the positive strain, after solubilization and activation with Proteinase K.

Results

Previous screening of Bt standard strains revealed the presence of a partial ps4 gene in Bacillus thuringiensis coreanensis strain. The alignment obtained by the BLAST tool revealed 100% identity between the fragment detected in this work with a hypothetical protein (ANN35810.1) from the genome of that same strain. Considering this, the isolation of the complete gene present in this strain was performed by applying the polymer chain reaction (PCR) technique, using the hypothetical sequence as a basis for the primers elaboration. The in silico analysis of the obtained sequence revealed 92.03% similarity with the ps4 sequence presented in the database (AB180980). Protein modeling studies and comparison of their structures revealed that the B. thuringiensis coreanensis has a new subclass of PS4, denominated PS4Ab1, being an important source of parasporin to be explored in biotechnological applications.

Introduction

Renowned for its broad insecticidal capacity, the bacterium Bacillus thuringiensis (Bt) produces endotoxin crystals that are lethal to various insect pest larvae (Oliveira-Santos et al., 2023; Rezaei, Moazamian & Montazeri-Najafabady, 2023). Bt is currently of significant agro-economic importance, widely employed for the control of agricultural pests and disease vectors of global relevance (Oliveira-Santos et al., 2023).

Found abundantly in nature, Bt exhibits extensive genetic diversity. Research has predominantly focused on its main insecticidal proteins, Cry and Vip (Mendoza-Almanza et al., 2020; Oliveira-Santos et al., 2023). Cry toxins, which are water-soluble, crystallize during the stationary phase of bacterial growth, forming protein inclusions in crystalline structures (Jallouli et al., 2020; Azizoglu et al., 2023). When ingested by target insects, these crystals dissolve in the alkaline midgut environment, releasing Cry toxins that bind to specific receptors on the midgut membrane, leading to pore formation, membrane disruption, and ultimately cell death through lysis (Jallouli et al., 2020; Azizoglu et al., 2023).

According to Oliveira-Santos et al. (2023), Bt is a multifunctional bacterium with considerable biotechnological potential. Beyond its insecticidal effects on insects, mites, protozoa, nematodes, and other invertebrates, it produces other proteins of biotechnological interest, such as parasporins (PS), which exhibit potent cytotoxic effects and non-hemolytic properties (Mendoza-Almanza et al., 2020; Oliveira-Santos et al., 2023).

Parasporins were first discovered by Mizuki et al. (1999) and are currently classified into six classes based on their structures and activity levels: PS1, PS2, PS3, PS4, PS5, and PS6. These proteins are cytotoxic to human cancer cells, primarily targeting liver cancer cells (HepG2), cervical cancer cells (HeLa and TCS), lung cancer cells (A549), colorectal cancer cells (Caco-2), endometrial adenocarcinoma cells (Sawano), and leukemic cells (MOLT-4, HL-60, and Jurkat) (Okassov et al., 2015; Santos et al., 2022). Like Cry toxins, parasporins require proteolytic activation for their functionality, which can be performed by enzymes such as trypsin or proteinase K, present in the tumor microenvironment or digestive system (Santos et al., 2022).

Initially, parasporins were identified as Cry proteins, even though they are non-insecticidal parasporal inclusion proteins with only 25% amino acid sequence similarity to Cry proteins (Ohba, Mizuki & Uemori, 2009). These molecules followed the nomenclature proposed by Crickmore et al. (1998). The term “parasporin” was introduced in 2006 after the formation of the “Committee of Parasporin Classification and Nomenclature” (CPCN), which remains responsible for the nomenclature of non-hemolytic bacterial parasporal proteins with selective cytotoxicity toward cancer cells.

Among the applications of parasporins, their potential in cancer therapy is particularly noteworthy. Cancer is a pressing global public health issue, with 21 types of cancer estimated to occur in Brazil alone (de Oliveira Santos et al., 2023). Current cancer treatments in Brazil include surgery, chemotherapy, radiotherapy, and bone marrow transplantation. Most cases require combining multiple modalities (Portal da Saúde SUS, Brazil, 2022). The search for alternative treatments that are both effective and less aggressive remains a challenge, and parasporins could represent a promising solution (Aboul-Soud et al., 2019).

Studies on the cytocidal effects of parasporins have analyzed the specificity of their receptors in mutated cells. This specificity may be attributed to the overexpression of receptors in cancer cells compared to normal cells, enabling selective tumor cell death (Okassov et al., 2015; Chubicka et al., 2018). Among the limited research conducted on these proteins, studies on the PS4 protein are particularly scarce. To date, only one subclass of PS4, described in the B. thuringiensis shandongiensis strain, has been characterized (Okumura et al., 2010).

Materials and Methods

Detection of parasporin 4 in Bacillus thuringiensis standard strains

Bt standard strains were obtained from the bank of standard strains and isolates at the Laboratory of Bacterial Genetics and Applied Biotechnology, Faculty of Agricultural and Veterinary Sciences (FCAV/UNESP), Jaboticabal, São Paulo, Brazil. These strains were screened for the presence of the ps4 gene using genomic DNA extracted following the method described by Bravo et al. (1998), with modifications replacing traditional water bath incubation with a dry-block. Gene detection was performed using polymer chain reaction (PCR) with primers and amplification conditions outlined by Chubicka et al. (2018), targeting a partial 736-bp fragment.

The ps4 gene fragments from positive Bt strains were cloned into a pGEM T-Easy vector (PROMEGA) according to the manufacturer’s instructions, and sequenced using a first-generation automatic sequencer (Sanger method) with 96 capillaries (ABI 3730 xl DNA Analyzer; Applied Biosystems, Foster City, CA, USA). In silico analyses were conducted using BioEdit, BLAST, and Clustal Omega to confirm the presence of the ps4 gene.

Isolation of the complete ps4 gene from Bacillus thuringiensis

The complete ps4 gene was isolated from DNA extracted from the selected strain, using the primer pair PS4F (5′-GGTGGATCATATGGCGATTA-3′) and PS4R (5′-CTACTGAAGAGATACAACTGGATCAAAG-3′), designed based on a hypothetical Bt gene sequence from GenBank (accession number CP016197.1:47747–48574).

The amplification reaction included 200 ng of genomic DNA, 10 pmol of each primer, 10 mM dNTPs, 50 mM MgCl2, and 0.3 U of Taq Polymerase in a suitable buffer. PCR conditions were as follows: initial denaturation at 94 °C for 2 min, followed by 30 cycles of denaturation at 94 °C for 1 min, annealing at 59 °C for 1 min, and extension at 72 °C for 1 min, with a final extension at 72 °C for 10 min. Amplified products were visualized on a 1% agarose gel by horizontal electrophoresis.

Cloning and sequencing of the ps4 gene from Bacillus thuringiensis

The product obtained by PCR referring to the complete ps4 gene was purified and cloned to confirm the gene by sequencing. Product purification was performed using the QIAquick PCR Purification Kit (QIAGEN) according to the manufacturer’s instructions. The DNA obtained was quantified via spectrophotometry using a Nanodrop device (ThermoScientific®, Waltham, MA, USA), and the integrity was confirmed on a 1% agarose gel. The purified amplicon was cloned into the pGEM T-Easy vector (Promega, Madison, WI, USA) following the manufacturer’s protocol.

Competent E. coli DH10B cells were transformed with the recombinant plasmid using the heat shock method (Sambrook & Russell, 2001). Transformant clones were cultured individually in 5 mL of Luria Bertani (LB) medium with ampicillin at 37 °C for 12 h with constant agitation at 200 rpm. Plasmid DNA was extracted via alkaline lysis with sodium dodecyl sulfate (SDS) (Sambrook & Russell, 2001), quantified using a spectrophotometer, and stored at −20 °C until further analysis.

PCR, using the ps4 gene-specific forward primer and the T7 reverse primer for the pGEM T-Easy vector, confirmed the presence and correct insertion of the gene. Amplified products were analyzed on a 1% agarose gel, and the sample with the confirmed amplicon was purified and sequenced using the Sanger method (ABI 3730 xl DNA Analyzer; Applied Biosystems, Waltham, MA, USA).

In silico analysis of the obtained sequence

The ps4 gene sequence was edited using BioEdit 7.7 (Hall, Biosciences & Carlsbad, 2011) and analyzed with BLAST (Altshul, 1990). The blastn and blastp algorithms were used to compare the sequence with parasporins in GenBank (Kuroda et al., 2013). Gene and protein alignments were performed using Clustal Omega (1.2.4) (Madeira et al., 2024) and visualized with Geneious Prime 2025.0 (Kearse et al., 2012).

Phylogenetic relationships were inferred using IQ-tree 2 (Minh et al., 2020) to construct an unrooted tree with 1,000 bootstrap replicates, visualized using iTOL v7 (Letunic & Bork, 2007). Protein structure predictions were conducted using AlphaFold2 (Jumper et al., 2021), with outputs analyzed for predicted alignment error (PAE) and predicted local distance difference test (pLDDT) values.

PAE is measured in Ångströms (Å), and for error analysis, values closer to 0 were considered to indicate higher reliability, while values closer to 30 indicate lower reliability (Varadi et al., 2022). Regarding pLDDT, residues with pLDDT < 50 are classified as having very low confidence; residues with 50 ≤ pLDDT < 70 have low confidence; residues with 70 ≤ pLDDT < 90 are considered confident; and residues with pLDDT ≥ 90 are classified as having very high confidence in the model (Varadi et al., 2022; Boland & Ayres, 2024).

Ramachandran plots were generated using PROCHECK (Laskowski et al., 1993) to assess stereochemical quality (Ramachandran, Ramakrishnan & Sasisekharan, 1963; Park et al., 2023). The number of residues in the most favored, additionally allowed, generously allowed, and disallowed regions was analyzed as important data.

AlphaFold2 generates a PDB file, which can be analyzed using standard molecular visualization systems such as PyMOL (Veit, Gadalla & Zhang, 2022). Thus, its visualization in cartoon form and the comparison between the protein modeled by AlphaFold2 and the 3D structure of the PS4 protein (2D42) from the Protein Data Bank (PDB) were conducted using PyMOL (DeLano, 2002). Furthermore, the InterPro platform (McDowall & Hunter, 2011) was used to analyze the domains of both PS4 proteins (Aktar et al., 2019).

Solubilization and activation of the PS4 protein from Bacillus thuringiensis protein crystals

Proteins were extracted from Bt strains cultured on nutrient agar (NA) (Gordon, Haynes & Pang, 1973) and in nutrient yeast extract salt medium (NYSM) (Yousten, 1984; Martins et al., 2007). After cultivation, bacterial cultures were centrifuged, and the pellet was washed twice with ice-cold sterile water, followed by centrifugation at 13,000 rpm for 10 min at 4 °C (Santos, 2018).

The pellet was dissolved in 1 mL of solubilization buffer (50 mM Na2CO3, pH 10; 1 mM EDTA; 10 mM DTT) and incubated for 1 h at 37 °C. After centrifugation, the supernatant was collected, and the pH was adjusted to 8 (Santos, 2018). The solubilized protein was filtered through a 0.45 µm membrane and the protein concentration was determined using the Bradford method with bovine serum albumin as the standard (Bradford, 1976).

PS4 has an inactive protein of 31–34 kDa, and when activated by proteinase K, it produces an active protein of 25–28 kDa (Suárez-Barrera et al., 2021; Santos et al., 2022; Rezaei, Moazamian & Montazeri-Najafabady, 2023). To analyze the strain’s protein profile, part of the sample was used for activation, while the remainder was frozen at −20 °C for application to a polyacrylamide gel.

Inactive PS4 protein (31–34 kDa) was activated by incubation with 30 µg/mL of Proteinase K at 37 °C for 1 h under low agitation. Activation was halted by adding phenylmethylsulfonyl fluoride (PMSF) to a final concentration of 1 mM (Rodrigues, 2012). Both protoxin and toxin forms were analyzed using sodium dodecyl sulfate-polyacrylamide gel electrophoresis (SDS-PAGE) (12%) (Laemmli & Favre, 1973).

Results

The screening analysis revealed that, among the standard strains assessed, only two tested positive for a partial fragment of PS4 (Fig. 1). One was the Bacillus thuringiensis shandongiensis strain, used as a positive control since it is described in the literature as carrying the ps4 gene (Okumura et al., 2008). The other was the Bacillus thuringiensis coreanensis strain, which had not yet been reported to carry this gene.

Figure 1 Electrophoresis in 1% agarose gel showing the amplified fragment of 736 bp of the ps4 gene from Bacillus thuringiensis coreanensis.

PC, Positive control represented by parasporin 4 fragment from Bacillus thuringiensis shandongiensis; BtcPS4, fragment of parasporin 4 found in Bacillus thuringiensis coreanensis (Indicated by the blue arrow). Ladder: 1 kb molecular marker (Fermentas); NC, negative control for the PCR reaction containing sterile Milli-Q water in place of the DNA.

Sequencing of the fragment confirmed, via the BLAST tool (blastn), a 100% identity with the sequence found in plasmid pST7-3, annotated as a “hypothetical protein.” This plasmid, fully sequenced and belonging to the B. thuringiensis coreanensis strain (ST7), has its complete genome deposited in GenBank (CP016197.1) (Zhu, J. Rice Research Institute, Sichuan Agricultural University, Huiming, Wenjiang, Sichuan 611130, China).

When compared to the only complete ps4 gene sequence available in the database for B. thuringiensis shandongiensis (AB180980), the partial fragment identified in this study exhibited 93% identity. Detection of the ps4 gene in the B. thuringiensis coreanensis strain facilitated further work, including the design of new primers based on the hypothetical sequence, enabling an in-depth analysis of PS4 presence in this strain.

Isolation of the ps4 gene from the hypothetical protein (CP016197.1:47747–48574) revealed non-conserved regions at the sequence’s start, complicating primer design for PCR reactions. To address this, the forward primer was synthesized to include 10 bases before the gene of interest, using the complete genome of plasmid ST7 from B. thuringiensis coreanensis available in GenBank (CP016197.1). This approach enabled amplification of the complete gene, with an expected size of 838 bp. The database gene measures 828 bp, and the additional 10 bases were accounted for in the forward primer design.

The PCR product from B. thuringiensis coreanensis was purified and cloned into the pGEM T-Easy vector. Cloning was confirmed via agarose gel electrophoresis, demonstrating the correct reading frame and promoter integration, yielding an expected value of 1,018 bp when considering the amplicon and vector sequences (Fig. 2).

Figure 2 Electrophoresis in 1% agarose gel showing the amplification product of the ps4 gene of Bacillus thuringiensis coreanensis, from the plasmid DNA of a positive clone of Escherichia coli DH10B.

Ladder: 1 kb molecular marker (Fermentas); 1: clone of the ps4 gene with a size of 1,018 bp (indicated by the blue arrow); NC, negative control for the PCR reaction containing sterile Milli-Q water in place of the DNA.

Analysis of the complete ps4 gene sequence using the blastn tool revealed 100% similarity to the hypothetical protein sequence (CP016197.1:47747–48574) and 92.03% similarity to the ps4Aa1 gene sequence in the database (AB180980). These results are visualized through gene alignments in the Geneious tool (Fig. 3).

Figure 3 Analysis of alignments of amino acid sequences of PS4 using the “Geneious” tool.

(A) Alignment of the sequence of the hypothetical protein gene (HProtein) from the database with the complete sequence of theps4gene (ps4Ab1) identified in this study, showing 100% similarity, as indicated by the completely green identity band. (B) Alignment of theps4Aa1sequence (AB180980.2) from the database with the complete sequence of the gene found (ps4Ab1), showing some dissimilarities, marked by white spots within the green identity band.

Protein analyses of PS4Aa1 (BAD22577.1) and the hypothetical protein (ANN35810.1) demonstrated a 90.55% similarity via the blastp tool. This similarity was further corroborated by amino acid sequence alignments using Clustal Omega (Fig. 4).

Figure 4 Alignment of the sequence of the hypothetical protein (ANN35810.1) with the only subclass described in the database, PS4Aa1 (BAD22577.1), using the “Clustal Omega” tool.

According to the blastp tool, both sequences share 90.55% similarity. Additionally, the hypothetical protein shows 100% similarity with the sequence identified in this study.

These findings suggest that the cloned sequence corresponds to a new subclass of parasporin 4, designated as ps4Ab1. The designation follows the parameters for Bacillus thuringiensis pesticidal protein nomenclature (Crickmore et al., 2021), as no specific nomenclature exists for parasporins. The new subclass displayed sequence similarity ranging from 76% to 96% with database sequences, warranting a classification change at the third level (Crickmore et al., 2021).

Sequence similarity was further confirmed in an unrooted phylogenetic tree (Fig. 5). Constructed using the maximum likelihood method, the tree aligns sequences and determines the most probable evolutionary model among possible configurations (Kapli, Yang & Telford, 2020).

Figure 5 The maximum likelihood phylogenetic tree, which includes “bootstrap” values for the different classes and subclasses of Parasporins, also incorporates the protein identified in this study (PS4Ab1).

This tree, built using the “IQTree” tool and edited on the iTOL platform, reveals that the protein, proposed as a new subclass of PS4 (shown as a red branch), is closely related to Parasporin 4 from the database (PS4Aa1). Both proteins form a clade with strong support (“bootstrap” = 100).

Protein modeling with AlphaFold2 highlighted strong consistency in predictions for models 1 and 2, evidenced by lower predicted alignment errors (PAE) (Fig. 6). The PAE, spanning 0 Å (blue) to 30 Å (red), indicated high confidence in domain packing and positioning. Good coverage with reference sequences was observed, particularly between amino acids 50 and 200, with average identity at approximately 90%. Terminal regions showed reduced identity (Fig. 6).

Figure 6 Confidence metrics for the predicted structure of PS4Ab1.

(A) Predicted aligned error (PAE) for all five models. (B) Coverage of PS4Ab1 (BAD22577.1) in the multiple sequence alignment (MSA) generated by “colabfold_search” with mmseqs2. (C) The predicted local distance difference test (pLDDT) for all five models. (D) The highest confidence structure prediction (Rank 1) of the PS4Ab1 protein modeled in AlphaFold2 and visualized in PyMOL in the “Cartoon” model. Model 1 exhibited PAE values closer to 0, indicating higher reliability, and average LDDT values above 80, with certain regions exceeding 90, suggesting high structural precision.

The predicted local distance difference test (pLDDT) plot across five models revealed that model 1 consistently achieved average pLDDT scores above 80, with specific regions exceeding 90, indicating high structural accuracy (Mariani et al., 2013). Consequently, model 1 was selected for further analysis (Fig. 6).

Ramachandran plots generated via PROCHECK for PS4Ab1 indicated residues distributed as follows: most favored (89.3%), additionally allowed (7.3%), generously allowed (3%), and disallowed regions (0.4%). These results validated the stereochemical quality of the predicted protein structure.

Structural comparisons between PS4Aa1 (2D42) and PS4Ab1, performed using PyMOL, reaffirmed their similarity at the structural level (Fig. 7). InterPro analysis revealed identical domains and structures, with neither protein exhibiting conserved blocks or typical Cry-3d insecticidal protein structures (Okumura et al., 2008).

Figure 7 Models visualized in the PyMOL tool in the “Cartoon” model.

(A) Structure of the PS4Aa1 (2D42) protein exported from the PDB database (“Protein Data Bank”) and visualized in PyMOL. (B) Structure of the PS4Ab1 protein modeled in AlphaFold2 and visualized in PyMOL. (C) Structure of PS4Aa1 (BAD22577.1) and PS4Ab1 proteins aligned in the PyMOL program to visualize their structural similarities. The partial overlap between the structures indicates that while the proteins are similar, they are not identical.

SDS-PAGE analysis of solubilized and activated PS4Ab1 from B. thuringiensis coreanensis protein crystals identified a band near 34 kDa, corresponding to inactive PS4, and a band at 27 kDa, representing active PS4 (Fig. 8).

Figure 8 Electrophoretic profile in SDS-PAGE gel 12% of the total solubilized protein of the Bacillus thuringiensis coreanensis strain before and after activation with Proteinase K.

1: “Spectra TM Multicolor Broad Range Protein Ladder” marker; 2: total non-activated solubilized protein showing the band corresponding to the non-active PS4Ab1 protein (indicated by the red arrow); 3: total protein solubilized and activated with Proteinase K, showing the band corresponding to the activated PS4Ab1 protein (indicated by the blue arrow).

Discussion

The analysis of the partial sequence of the ps4 gene identified in the B. thuringiensis coreanensis strain during screening suggested that the ps4 gene is likely located on plasmid pST7-3 or a similar plasmid, a hypothesis confirmed by complete gene sequencing. The protein, previously described as hypothetical by Zhu (CP016197.1), encoded by the nucleotide sequence present in plasmid pST7-3, represents a parasporin of the PS4 class.

The detection of another standard strain carrying PS4 highlights the need to extend screenings to other Bt strains and isolates. While standard strains are often recognized for their larvicidal or hemolytic efficiency (Mizuki et al., 2000; Kitada et al., 2006; Akiba & Okumura, 2017), the current understanding of parasporin protein toxicity and their dual insecticidal and cytotoxic roles remains limited.

Studies indicate that parasporins may exhibit both activities, with their presence not restricted to non-insecticidal strains (Soberón et al., 2018; Mendoza-Almanza et al., 2020; Santos et al., 2022). The differentiation between Bt strains with antitumoral activity and those without remains an area requiring further exploration.

Previously, B. thuringiensis coreanensis was reported to exhibit efficacy against certain cancer cell lines, with its parasporal crystalline proteins described as non-insecticidal and non-hemolytic (Namba et al., 2003). However, no prior in silico study has examined the hypothetical protein in this strain that shows significant similarity to PS4 in the database.

The genome of B. thuringiensis coreanensis includes plasmid ST7, which carries five genes for insecticidal crystalline proteins: Cry22, Cry32, Cry45, Cry62, and Cry73 (Zhang et al., 2020). The PS4Aa1 protein, formerly known as Cry45Aa1, exhibits homology with Cry45 present in plasmid ST7, suggesting that both may have similar toxicity profiles toward cancer cells. Therefore, the parasporin identified in this study may yield comparable results given its plasmid association.

The constructed phylogenetic tree demonstrated high bootstrap statistical support across most internal nodes, confirming the proximity between parasporin classes and the relationship between PS4Aa1 and PS4Ab1, which share the same node and are considered sister groups with strong clade support (100) (Wiley & Lieberman, 2011).

Clustal Omega alignments highlighted sequence similarities used to construct the phylogenetic tree. This method aligns the most similar sequence pairs first, progressively adding more distantly related sequences according to the guide tree (Kapli, Yang & Telford, 2020). The tree indicates that, aside from its similarity to PS4Aa1, PS4Ab1 is also closely related to parasporins 2, 3, and 5.

The proximity between PS2 and PS4 classes extends beyond this phylogenetic tree. Both parasporins target similar cancer cell lines and share functional similarity through aerolysins, which form β-type pores. These structures bind non-specifically to cell plasma membranes, forming oligomers that induce target cell death (Akiba et al., 2009; Ohba, Mizuki & Uemori, 2009).

InterPro analysis reveals that both PS4 proteins belong to the aerolysin superfamily, which includes bacterial toxins such as Clostridium perfringens epsilon toxin (ETX) and Bacillus MTX2 mosquitocidal toxin. These act as cytolysins by forming pores in target membranes (Girija et al., 2023).

PS4 is a β-pore-forming toxin with cholesterol-independent activity, damaging plasma membranes in susceptible cells (Girija et al., 2023). According to the literature, PS4Aa1 (BAD22577.1) shares sequence similarities with epsilon toxin (ETX) (21%) and aerolysin toxin (10%) (Velasquez, Rojas & Cerón, 2018). These toxins bind to eukaryotic cells, aggregating to form pores in lipid bilayers, leading to membrane permeability disruption and osmotic lysis (Aktar et al., 2019).

Thus, both PS4Aa1 and PS4Ab1 likely exhibit similar functionalities, binding non-specifically to plasma membranes to form pores in target cells (Velasquez, Rojas & Cerón, 2018). However, the actual action of PS4Ab1 requires confirmation through cytotoxicity assays.

The solubilization and activation of PS4Ab1 from B. thuringiensis coreanensis protein crystals, along with SDS-PAGE analysis, revealed expected band sizes for both inactive (34 kDa) and active (27 kDa) protein forms, consistent with literature reports (Suárez-Barrera et al., 2021; Santos et al., 2022; Rezaei, Moazamian & Montazeri-Najafabady, 2023). These findings further underscore the similarity between PS4Ab1 and PS4Aa1 (BAD22577.1).

Literature indicates that Bt strains presenting non-insecticidal Cry proteins are more abundant in nature than those with insecticidal Cry proteins (Santos et al., 2022; Akiba & Okumura, 2017; Ohba & Aizawa, 1986). This knowledge led to the discovery of parasporins, and the identification of this new subclass reaffirms the potential for discovering additional parasporin subclasses or even entirely new classes in the future.

Conclusions

Screening of Bacillus thuringiensis standard strains identified B. thuringiensis coreanensis as a new source of parasporin 4, where a protein previously annotated as hypothetical (ANN35810.1) in its genome demonstrated high similarity to PS4 from the database (AB180980).

The sequence of this hypothetical protein was characterized as a new subclass of PS4 protein, named PS4Ab1. In silico analysis of parasporin PS4Ab1 proposed a model for this protein, highlighting key structural elements relevant to its cytotoxic activity against tumoral eukaryotic cells. This protein may exhibit similar functionality to parasporin PS4Aa1, suggesting its potential application in human and animal medicine.

Future cytotoxicity assays will be essential to confirm the precise action of this protein on cancer cells and to reaffirm its classification within the parasporin family.

Although Bacillus thuringiensis strains and isolates continue to be of significant scientific interest, parasporins remain an underexplored area of research. This study aims to advance the understanding of these proteins and pave the way for further investigations.

Supplemental Information

Supplemental Information 1 Partial sequence found in the first PCR performed with the positive strains.

Supplemental Information 2 Complete sequence of the ps4Ab1 gene.

Supplemental Information 3 List of Bacillus thuringiensis strains used for the screening.

Supplemental Information 4 Amino acid sequences used to construct the phylogenetic tree.

Supplemental Information 5 PS4Ab1 protein modeling files.

These data were obtained from modeling the PS4Ab1 protein using the Alphafold2 tool, its files can be opened in Pymol, Notepad and common image viewer.

Supplemental Information 6 PS4Aa1 protein PDB file obtained from the protein database.

Supplemental Information 7 Ramachandran files for all models.

The authors would like to thank Dr. Saura Rodrigues da Silva, Dr. Elisângela Soares Gomes Pepe, and biologist João Victor dos Anjos Almeida from the Faculty of Agricultural and Veterinary Sciences (FCAV/UNESP), Jaboticabal, São Paulo, Brazil, for their collaboration and assistance with phylogenetic analysis, Bradford analysis, and structural bioinformatics analysis, respectively. The authors used ChatGPT, an artificial intelligence tool developed by OpenAI, for linguistic revision of this manuscript.

Additional Information and Declarations

Competing Interests

The authors declare that they have no competing interests.

Author Contributions

Thais N. F. Santos conceived and designed the experiments, performed the experiments, analyzed the data, prepared figures and/or tables, authored or reviewed drafts of the article, and approved the final draft.

Raquel O. Moreira performed the experiments, analyzed the data, authored or reviewed drafts of the article, and approved the final draft.

Jardel D. B. Rodrigues analyzed the data, authored or reviewed drafts of the article, and approved the final draft.

Luis A. C. Rojas analyzed the data, prepared figures and/or tables, and approved the final draft.

Jackson A. M. Souza analyzed the data, authored or reviewed drafts of the article, and approved the final draft.

Janete A. Desidério conceived and designed the experiments, analyzed the data, authored or reviewed drafts of the article, and approved the final draft.

DNA Deposition

The following information was supplied regarding the deposition of DNA sequences:

The PS4Ab1 sequence is available at GenBank: PP524979.1.

Data Availability

The following information was supplied regarding data availability:

Raw data are available in the Supplemental Files.

The gromacs files of stability tests of the proposed models using at least 100-ns long molecular dynamics (MD) simulations are available at Zenodo: Santos, T. (2024). Molecular dynamics of the PS4Ab1 [Data set]. Zenodo. https://doi.org/10.5281/zenodo.13905386.

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
