# Peer review of "Isolation and in silico analysis of a new subclass of parasporin 4 from Bacillus thuringiensis coreanensis"

_PeerJ, doi:10.7717/peerj.19061_

## Round 0.1 · original submission · Major Revisions

· Academic Editor

Major Revisions

Please address the comments of both reviewers in a point wise manner.

Reviewer 1 ·

Basic reporting

Santos et al. report an investigation into the parasporin 4 (PS4) protein in Bacillus thuringiensis coreanensis, identifying a new subclass, PS4Ab1. The main text of the manuscript is clearly written, albeit the figure captions need some revision for clarity. I suggest extending the introduction, particularly regarding the current narrative concerning the potential of parasporins.

It is nice to mention the main interest in Bacillus thuringiensis, which is insect pest control. However, I found this part quite limited in scope. I believe a brief discussion regarding the well-characterized mode of action of Cry toxins will benefit the manuscript as it could give a contrast for the less characterized parasporins. For instance, see https://www.sciencedirect.com/science/article/abs/pii/S0048357512000697, and https://www.sciencedirect.com/science/article/abs/pii/S0965174824000043.

Please also add relevant citations for the claims in the introduction e.g., for "the bacteria Bacillus thuringiensis (Bt) presents endotoxic crystals that are lethal to different insect pest larvae". Please also expand on the potential biotechnological applications of parasporins with further citations.

Experimental design

The in silico and experimental methodologies are appropriate for the study and sufficiently described, albeit I found them to be limited in scope.

Please add the versions of the in silico tools and provide references (or GitHub/website links) where possible.

Validity of the findings

I think the claims regarding the potential biotechnological applications of parasporins should be toned down in the manuscript as there is no direct data supporting this claim for the characterized parasporin. Especially, I found the "its potential biotechnological application" in the title quite out of place for this manuscript. This could be a minor point in the discussion, but including in the title does not seem appropriate.

Additional comments

-Fig 6: Please provide the definition of "pLDDT" in the caption.
-Fig 7C: It will help to indicate the directions of the rotations of 3D models with arrows in the figure.

·

Basic reporting

This study, which revealed the presence of a partial ps4 gene in the Bacillus thuringiensis coreanensis strain, is important in terms of revealing a new group of parasporins known to have cytotoxic effects. However, although it was not done in the study, the continuous mention of the cytotoxic effects of this new parasporin distorts the purpose of the study. The study reflects that the cytotoxic effects of the newly discovered parasporin PS4Ab1 against cancer cells have been investigated. This situation does not match the content of the study. As the authors also mentioned, information should be provided about the activities of the newly discovered parasporin after its cytotoxic effects against cancer cells have been investigated. Therefore, it would be appropriate to revise the Discussion section.

Experimental design

There is not enough explanatory information about the cloning and sequencing of the ps4 gene in the material method section. The method section should be written more clearly.

In silico analysis of the Ps4 gene and how its 3D structure was extracted should be explained in more detail.

“The amplification conditions were initial denaturation at 72ºC for 2 min, 30 cycles of denaturation at 72ºC for 1 min, annealing at 59ºC for 1 min, extension at 94ºC for 1 min, and at the end of the cycles an extension at 94ºC for 10 min.” This sentence should be verified for accuracy. Are the denaturation step and extension steps correct?

Validity of the findings

The discovery of a new subtype of parasporins, known to have cytotoxic effects especially against cancer cells, is scientifically important. However, the possible cytotoxic effects of this new type need to be proven experimentally. The effective cytotoxic effects of previously known ps4 parasporins provide the expectation that this newly discovered parasporin PS4Ab1 may also have a possible cytotoxic effect. Therefore, the findings obtained and those that may be obtained in the future have the potential to make significant contributions to the scientific world.

Additional comments

No additional comments.

---

## Round 0.2 · accepted · Accept

· Academic Editor

Accept

Authors have addressed all of the reviewers' comments and the manuscript is ready for publication.

Reviewer 1 ·

Basic reporting

The revisions have successfully addressed the authors' concerns, and I have no further comments.

Experimental design

no comment

Validity of the findings

no comment

Additional comments

no comment